# Knowledge Production for Resilient Landscapes: Experiences from Multi-Stakeholder Dialogues on Water, Food, Forests, and Landscapes

**Anna Tengberg \*, Malin Gustafsson, Lotta Samuelson** 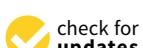 **and Elin Weyler** 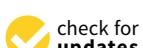

Stockholm International Water Institute, Swedish Water House, Box 101 87, 100 55 Stockholm, Sweden; malin.gustafsson@siwi.org (M.G.); lotta.samuelson@siwi.org (L.S.); elin.weyler@siwi.org (E.W.)
\* Correspondence: anna.tengberg@siwi.org; Tel.: +46-760-060406

**Abstract:** Landscape-wide approaches integrating agriculture, forestry, energy, and water are considered key to address complex environmental problems and to avoid trade-offs. The objective of this paper is to analyse how knowledge production through multi-stakeholder dialogues on water, landscapes, forests, and agriculture can inform governance and the management of landscapes. Multi-stakeholder learning dialogues and platforms (MSPs) were established related to water and natural resources management, complemented by targeted reviews, to establish a shared understanding of the drivers of change and impacts on the hydrology of landscapes and ecosystem services. The MSP dialogues illustrate the need to address water as an integral part of landscape management and governance to achieve the wide range of the Sustainable Development Goals related to water and food security, climate action, life on land, as well as sustainable production and consumption, equality, and strong institutions. The co-production of knowledge through MSPs contributes to continuous learning that informs adaptive management of water flows in landscapes, above and below ground, as well as in the atmosphere. It helps to build a shared understanding of system dynamics and integrate knowledge about hydrology and water flows into policy recommendations. Co-production of knowledge also contributes to stakeholder participation at different levels, inclusiveness, and transparency, and to water stewardship.

**Keywords:** multi-stakeholder platforms; ecosystem services; landscapes; water governance

## 1. Introduction

Landscape-wide approaches that integrate the development of agriculture, forestry, energy, and water are considered key to address complex environmental problems [1,2], and to avoid trade-offs between response options. Applying the landscape approach is particularly useful when integrated solutions are required to solve complex challenges related to sustainable development [3]. Moreover, landscape approaches can be a mechanism for dialogue and discussion among multiple stakeholders regarding trade-offs to mobilise better land use and water resource outcomes [4]. The Convention on Biological Diversity (CBD) has 10 adopted principles for a landscape approach to reconcile agriculture, conservation, and other competing land uses. These principles emphasise the importance of multiple scales, multifunctionality, multi-stakeholder participation, resilience, and adaptive management [5]. Landscape degradation poses serious challenges to water and food security, biodiversity, and ecosystems, and for the ability of farmers and local communities to adapt to the impacts of climate change [6,7]. The process of land degradation also increases competition for natural resources and threatens people's livelihoods, well-being, food, water and energy security, as well as the resilience capacity of people and natural ecosystems. Forests and trees have key functions in maintaining resilient and productive landscapes, communities, and ecosystems. They ensure a water supply and provide

high quality water resources. In fact, around 75% of the world's accessible freshwater for agricultural, domestic, urban, industrial, and environmental uses depend on forests [8].

Sustainable development requires a balanced production of knowledge [9] from multiple sources to meet the needs of society more effectively and to inform sustainable policy directions [10]. Co-production is a complex concept and we therefore refrain from entering the discussion about its definitions and adopt a focus on co-production that informs the capacity to link knowledge with action in pursuit of sustainability [11]. Against this background, the objective of this paper is to analyse how knowledge production through multi-stakeholder dialogues on water, landscapes, forests, and agriculture can inform governance and management of landscapes. Multi-stakeholder learning dialogues and platforms (MSPs) on a series of themes related to water and natural resources management, complemented by targeted reviews, are used to establish a shared understanding of the drivers of change and impacts on hydrology of landscapes and ecosystem services, and to generate policy recommendations for action. Governance functions and attributes required to achieve synergies across sectors and scales related to water management in landscapes are discussed. Implications for achieving the Sustainable Development Goals (SDGs) underpinned by water concerns are also addressed, as well as commitments linked to the Multilateral Environmental Agreements (MEAs), such as the UN Framework Convention on Climate Change (UNFCCC).

## 2. Materials and Methods

### 2.1. Multi-Stakeholder Platforms for Co-Production of Knowledge

This paper builds on work by Swedish Water House (SWH), a part of the Stockholm International Water Institute (SIWI), that for over a decade has engaged in co-production of knowledge on the role of water in sustainable development, engaging practitioners at the national and local level, scientists, the private sector, and policymakers. The aim is to develop a common understanding of water challenges and clarify actions through shared insights in an interactive process. Most multi-stakeholder processes involve a preparation period in which stakeholders define their positions in relation to the issues, discuss, analyse, and share perspectives and then co-produce, e.g., a policy report, or policy-making initiative. The five-step process of initiation, mapping of key issues and actors, dialogue preparation, realisation and follow-up is typically overseen by a so called facilitating agent, such as SWH, who are responsible for convening and guiding the dialogue to ensure all voices are heard, thus having a critical role for the trust building process between stakeholders [12].

The MSPs facilitated by SWH are described as "networking initiatives for Swedish organisations to develop competence in water management and their relevance for a variety of fields such as economics, environment, gender, climate science, governance, ecosystem management and sanitation". SWH MSPs are small, manageable interdisciplinary networks that meet in person at SIWI in Stockholm on a regular basis, but some stakeholders participate via a video link if based elsewhere. The MSPs link up with ongoing international processes and networks through international meetings such as the World Water Week annual water conference in Stockholm, but the MSPs have also participated in meetings at other global fora outside of Sweden linked, e.g., to the MEAs. SWH creates the summaries of the discussions that are reviewed and validated by all MSP members. The evolution of the model is described in Figure 1. Findings from the MSPs discussed in this paper include: (1) Water and Food (2014–2017); (2) Water and Forest (2014–2016); and (3) Water and Landscapes (2017–2019). For a summary of published outputs see Table 1.

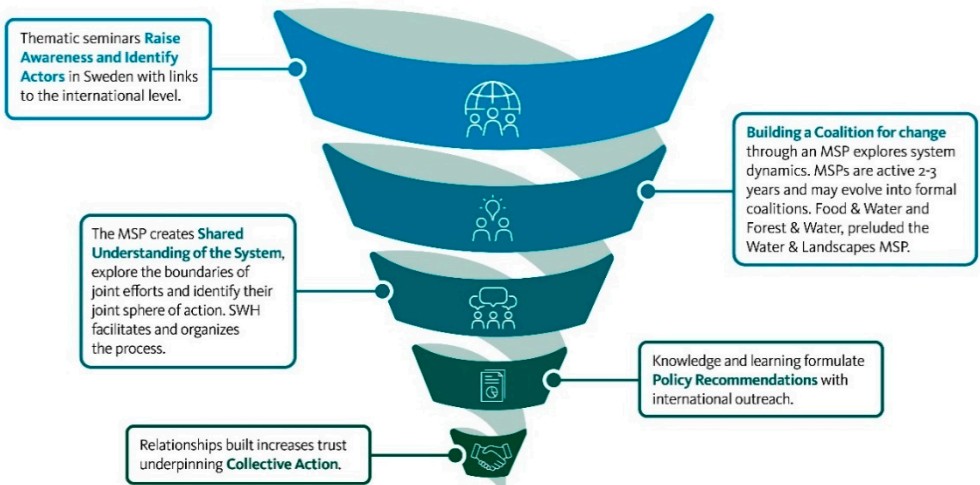

**Figure 1.** Establishing and sustaining multi-stakeholder learning dialogues and platforms (MSPs) over time related to the landscape theme (SIWI—Stockholm International Water Institute; SWH—Swedish Water House, WWW—World Water Week).

**Table 1.** Publications and online tools produced by SWH MSPs.

| | | |
|---|---|---|
| **MSP Reports** | Water for productive and multifunctional landscapes [13] | 2018 |
| | Water, forests, people—building resilient landscapes [14] | 2015 |
| | Adapting Water Management to Climate Change [15] | 2009 |
| | Agriculture, Water and Ecosystems. Swedish Water House [16] | 2007 |
| **Policy Briefs** | Managing the Forest-Water Nexus: Opportunities for Climate Change Mitigation and Adaptation | 2019 |
| | Agroforestry for adaptation and mitigation to climate change | 2019 |
| | How landscapes and water mitigate climate change | 2018 |
| | Water for productive and multifunctional landscapes | 2018 |
| | Championing the Forest-Water Nexus: Report on the meeting of key forest and water stakeholders | 2018 |
| | Water, forests, people—building resilient landscapes | 2015 |
| **Scientific papers** | Gaps in science, policy, and practice in the forest-water nexus [17] | 2019 |
| | Water, Forests, People: The Swedish Experience in Building Resilient Landscapes [1] | 2018 |
| **Others** | Publications under the Ethiopia Water and Landscape Governance Programme | 2020 |
| | Sweden's Government Agency for Development Cooperation (Sida) International Training Program 2021–2025: Locally Controlled Forest Restoration—LoCoFoRest A Governance and Market Oriented Approach for Resilient Landscapes | 2020 |
| | Water tools results report to Sida and the involved companies | 2018 |
| | Contribution by the Forest-Water Champions to the Talanoa Dialogue | 2018 |
| | Water, food, and human dignity—a nutrition perspective [18] | 2015 |
| | Water journey | 2015 |

Two early MSPs were precursors to the MSPs discussed in this paper: The Resilience and Freshwater Initiative [15], and the Swedish Network for the Comprehensive Assessment of Water Management in Agriculture [16]. These stressed the need to manage ecosystems on a catchment scale and the crucial role of stakeholder participation at multiple levels. The SWH MSP on Water and Forests followed, engaging professionals from the forest and water resource sectors in Sweden, with representation from forest authorities, universities, and other research organisations, industry, consultancy companies, smallholder organisations, civil society organisations, including environmental organisations

and the Church of Sweden (also a large forest/landowner). In total, more than 100 people from 42 academic, public and private sector, and civil society organisations participated in the process [14]. Collectively it was concluded that an integrated landscape approach involving a broad array of sectors and stakeholders is needed to achieve sustainable forest and water management.

This led to the establishment of an SWH MSP on Water and Landscapes that set out to answer questions about (1) how hydrology affects the productivity of the landscape and what hydrological aspects need to be considered when rehabilitating/restoring a landscape for sustainable production of nutritious food and other natural resources that contribute to sustainable growth locally, regionally and globally; and (2) which governance arrangements and management approaches enable and support the productivity of the landscape, minimise the risk of over-exploitation of water, and enables agreements between different stakeholders. Representatives from forest, agriculture, environment, water, and industry sectors participated, as did civil society, scientific institutions, and competent authorities, mainly from Sweden but also from international organisations. It included a total of 23 institutions. This MSP broadened the dialogue to cover other land uses in the landscape than forest, particularly agricultural land, as well as impacts of climate change [13].

In parallel, an SWH MSP on Water and Food focused on private sector engagement used a different approach to build trust and promote joint action among stakeholders. Closed workshops engaged two of the largest retailers in Sweden and some of their Sweden-based suppliers, the alcohol monopoly, as well one of the largest agricultural cooperatives in northern Europe. The aim was to have representation from the whole value-chain for food and beverages. The most commonly used international sustainability tools, such as certifications, standards or guidelines, were analysed or tested by the group to develop common recommendations about which tools most holistically addressed sustainable water use and management. The results were presented in a guide [19] together with a suggested approach for which tools to use at each stage of managing the water impact from food and beverage production.

The general theory of change for the MSPs used by SWH is in line with the resilience-based process developed by Enfors-Kautsky et al. (2018) [20]. The process suggested is very similar to the empirical approach that SWH has developed. Based on these two approaches, Figure 1 illustrates the theory of change and pathway from the establishment of an MSP, to the shared understanding of a system and its boundaries, to collective action.

Figure 2 shows the different steps in the theory of change (1–5) in grey circles, the boundary partners and stakeholders involved in the process, and their interlinkages. The MSP on Water and Food is displayed to the left, and the MSP for Water and Forests to the right, with links to the Water and Landscape MSP at the bottom that brings together inputs from all the different boundary partners. Figure 2 also maps the different knowledge sources used, from small-scale farmers and forest owners, to the private sector and academia, with links to international networks. Important boundary partners from both public and private sectors included farmer and forest owners' associations, civil society organisations (CSOs) and donors, such as the Sweden's Government Agency for Development Cooperation (Sida).

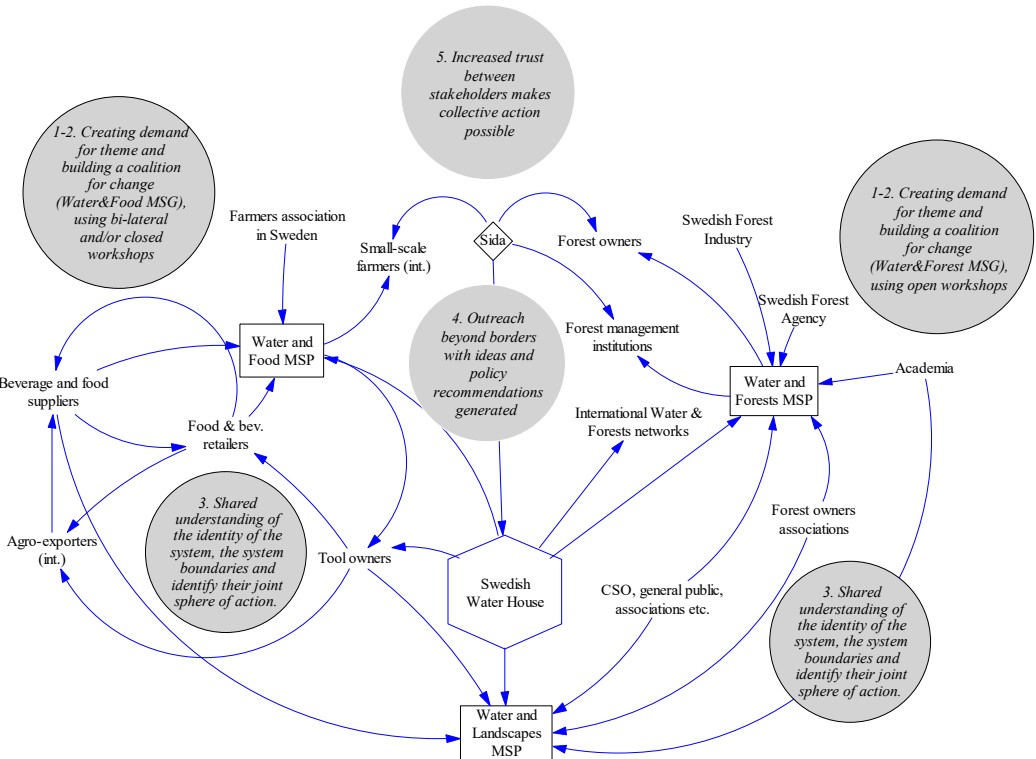

**Figure 2.** The approach for building multi-stakeholder platforms for dialogues on water, food, forests and landscapes with key boundary partners and stakeholders involved in the process and their interlinkages.

This paper draws on the dialogues and findings from the different SWH MSPs summarised in reports, policy briefs, synthesis of online tools and scientific publications (Table 1). These are complemented by findings from an international platform called the Forest-Water Champions (FWC) established in August 2017 by SWH, SIWI, the Food and Agriculture Organisation of the United Nations (FAO), and the International Union for the Conservation of Nature (IUCN). The objective of the FWC is to draw attention to the important role played by the forest-water nexus to reach the targets in Agenda 2030 and the Paris Agreement. The expert group meets annually to identify and communicate recommendations to policy makers.

### 2.2. Targeted Reviews

Two targeted reviews were conducted to complement and validate the MSP findings: (1) water-related ecosystem services; and (2) (Intended) Nationally Determined Contributions ((I)NDCs of the UNFCCC. The first targeted review was conducted based on scientific literature and reports relevant to ecosystem services. The aim was to identify water-related ecosystem services provided by forests and landscapes, and to highlight underlying ecosystem processes. A keyword search was undertaken using the keywords "ecosystem service(s)" and "service(s)". The list of water-related ecosystem services was then used in a scientific literature search including a synthesis of both qualitative and quantitative research to map underlying ecosystem processes.

In addition, the Forest-Water Champions network performed an analysis of reviews of (I)NDCs published between the years 2015–2019 to investigate the extent of forest-water synergies and interlinkages captured in these reviews. The aim of the synthesis was to identify to what extent the (I)NDCs pay attention to forest-water interlinkages, resilience and landscape approaches. The collection of (I)NDC reviews (Table S1) were analysed using a combination of table-of-content screening and advanced keyword searches, based on a set of keywords relevant for the forest-water nexus, i.e., forest, water, landscape,

land use, resilience/resilient and restoration. The extent of forest-water synergies and interlinkages identified in the reviews were examined based on the keywords, as well as the contexts of the text where the words were included.

The combined knowledge generated by MSPs and the targeted reviews was thus used in the subsequent conceptualisation and analysis of the role of water in landscapes.

## 3. Results

### 3.1. Conceptualising the Role of Water in Landscapes

The MSP on Water and Forests used the framework developed by the Millennium Ecosystem Assessment (MA) to identify the drivers of change of hydrology and water flows in landscapes and impacts on ecosystem services [14]. We combine the landscape approach with the MA conceptual framework [21], while also building on the findings of the IPCC (2019) [22] and IPBES (2018) [2] about the drivers of land degradation and loss of nature's contributions to people and sustainable development through water-related ecosystem services.

In a changing environment, the drivers of change of water flows in landscapes will impact ecosystem services. Future management of landscapes will be influenced by how these drivers of change affect the water cycle and water flows. Water management is critical for addressing tipping points, such as deforestation and land degradation [23]. Findings from the MSPs and targeted reviews are summarised below under each quadrant of the conceptual framework of Figure 3, starting with the drivers of change (A) and moving clockwise in the figure to B, C and D.

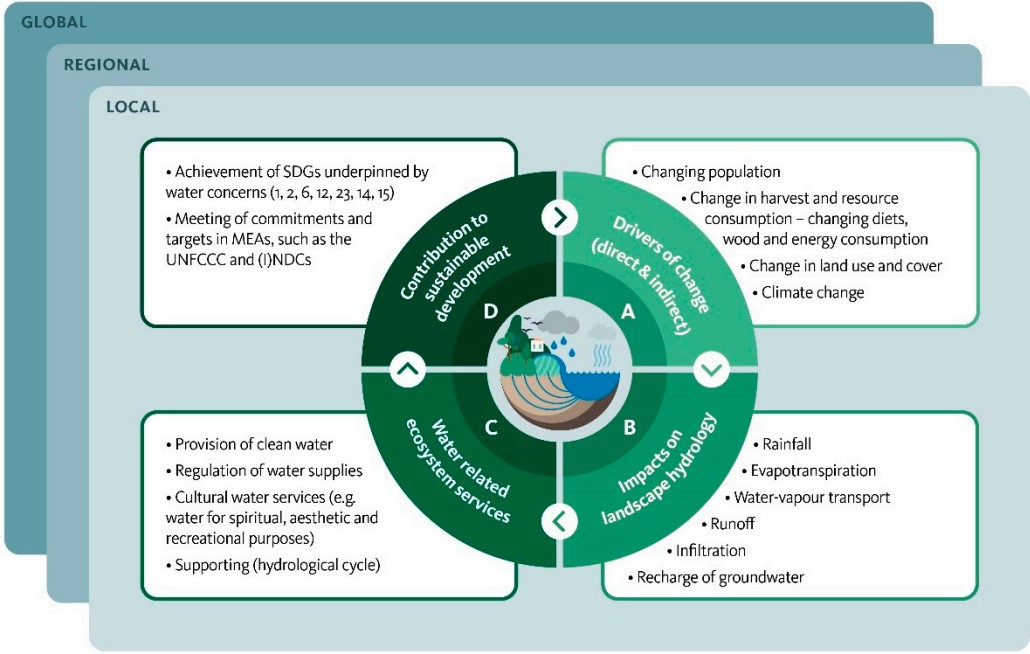

**Figure 3.** Conceptual framework for analysis of drivers of change, impacts on landscape hydrology and water-related ecosystem services and their contribution to sustainable development. (SDGs—Sustainable Development Goals; MEAs—Multilateral Environmental Agreements; UNFCCC-UN Framework Convention on Climate Change; (I)NDCs—(Intended) Nationally Determined Contributions).

### 3.2. Drivers of Change (A)

Changes in population and diets, wood, and energy consumption: At the global level, two key mega-processes are changing water use and water availability; and changing diets and increased population [24]. Worldwide, agriculture uses an average of 70 percent of all freshwater withdrawals, rising to 90 percent in many poorer countries [6]. Population

growth is estimated to increase food demand by 50 percent by the year 2050, while the area used for agriculture, as well as water withdrawals, is already reaching the desired limit for sustainable use globally [25]. There is a growing global demand for more meat, dairy, and eggs, as well as for fruit and vegetables. There is a challenge to satisfy the total demand for water due to increased population, as well as an increase in water use for more water demanding crops and livestock produce [18]. In addition, demand for wood raw material is expected to grow considerably, both in developed and developing countries. UN Habitat predicts that by 2025, another 1.6 billion people will require adequate, affordable housing. The current expansion of cities, housing needs and increasing household income boost the demand for wood-based products such as furniture, flooring, doors, etc. There is also an ongoing technology shift with new building technology where laminated wood has the potential to replace steel and concrete. This new and speedy development in high-income countries is a driver of the biobased economy also in current low- and mid-income countries where much of the housing expansion will happen [26].

Impacts of land use change on water flows: Human activities have transformed the earth surface by using both water and land worldwide [27]. Bio-geophysical properties linked to vegetation on land control evapotranspiration is one of the largest fluxes of water. This means that vegetation changes may change these intensive and extensive properties. The evapotranspiration can change magnitude and direction depending on both original and resulting vegetation cover after land transformation. At the global scale, human activities have already left a footprint on the freshwater system. Flow regulation has decreased the intra-annual variability of runoff due to the impoundment of water for hydropower production, homogenising runoff at the annual scale [28,29]. The wide-scale effect of irrigation and flow regulation appear also to be related to the increase in relative evapotranspiration observed in the largest 100 basins during the last 100 years [30].

Impacts of climate change: Climate change will alter water availability which will affect the capacity for healthy and productive landscapes to provide food, fodder, wood, and fibre. For example, in most parts of the world, seasonal temperatures are increasing, with more heatwaves, affecting crop water demand. In parts of Africa, there are a growing body of examples suggesting changes in both seasons and rainfall intensities [31]. There is a scientific consensus that expected (and confirmed) temperature increases and changes in temperature patterns may already be showing in some landscapes. In many locations, especially in the tropics and semi-arid areas, this may result in lowered yields due to heat stress and may also require more water for irrigation. The impact of climate change on hydrology at the landscape scale is still uncertain and highly unpredictable. In most cases, however, there may already be a new reality; more extreme events and greater occurrence of both drier and wetter conditions. This can impact multiple landscape features such as erosion, vegetation establishment and water availability (notably scarcity) for many people, with decreased food security as a result. According to the IPCC Climate Change and Land report [22], increases in water demand and water scarcity can be expected under all future socio-economic scenarios.

### 3.3. Impacts on Landscape Hydrology (B)

Forest, water, recycling ratios and hydrologic space: Trees and forests moderate water budgets, clean water, store carbon, enhance biodiversity and reduce erosion and runoff from landscapes. However, their impacts on the hydrological cycle at different scales are still poorly understood [17,32]. Research has shown the importance of looking outside the basin to understand how water is transported across continental and terrestrial surfaces, although conventional definitions of water balance are typically bound by the catchment [33]. It is important to consider the hydrological space and recycling ratios and identify what share of rainfall comes from recycled conventional evapotranspiration (ET). ET feeds an important share of terrestrial precipitation, and on average, forests provide more evapotranspiration (atmospheric moisture) than other land cover surfaces [34]. The large-scale spatial organisation and connectivity of land-use practices and forest cover must thus be



considered when addressing issues of forest cover, water availability and the hydrological cycle [32].

Optimum tree cover: A review on the impacts of afforestation and agroforestry on infiltration capacity in the tropics showed an increase of two to five times with trees [35]. Other studies have shown that in landscapes with scattered trees, soil infiltration capacity increases in the vicinity of trees as far as 20 m away from the closest tree stem. For instance, in systems with an open tree cover, such as agroforestry parklands or open woodlands, it is important to consider the water balance both in areas under trees, and in small and large gaps among trees [36]. Better soil structure under trees improves infiltration capacity and reduces surface runoff. That is, a higher percentage of water on the soil surface will absorb into the soil and thus be available for groundwater recharge. The specific tree density that maximises groundwater recharge will depend on several factors including climate, soil characteristics, tree species, tree age and size, tree spatial distribution, and land use and management.

Agriculture, water, and food production: Agriculture has contributed to a global redistribution of the spatial pattern of evapotranspiration, with decreasing ET in areas of large-scale deforestation and increasing ET in many irrigated areas with impacts on climate and ecosystems [37,38]. Agriculture often increases provisioning ecosystem services, while reducing other ecosystem services with effects on aquatic ecosystems, coastal zones, and wetlands, as well as terrestrial ecosystems, increasing the risks of crossing tipping points with negative feedback on agricultural production, food security and poverty reduction [21,39]. Several very important food-producing landscapes of the world are under threat of degradation, water insecurity and climatic change that may reduce productivity significantly [22,23].

*3.4. Water-Related Ecosystem Services (C)*

Landscapes and ecosystems are influenced heavily by the water cycle, and water is a crucial component in most, possibly all, ecosystem services. There is a reciprocal influence between forests, grasslands, soils, wetlands, and water [32]. Wetlands have particularly visible hydrological functions, such as the ability to store water, thereby helping to regulate floods [40,41]. There are many interactions between drivers of change and water-related ecosystem services. Water is inherently intertwined with all processes of social-ecological systems and can be both a control, state and driving variable [23], which makes it challenging to distinguish between the role of water as a driver of change, source of resilience and generator of ecosystem services [42]. The landscape concept is rarely used in these studies, except for studies on cultural ecosystem services, where landscapes and their water resources are considered important for heritage values and identity [43]. Nevertheless, the ecosystem service concept was useful in forming a shared understanding among diverse stakeholders of the need for collective action to avoid loss of water-related services important for the productive use of landscapes. Table 2 summarises the water-related ecosystem services important for sustainable development and the resilience of both landscapes and people, ranging from supporting, provisioning, and regulating to cultural ecosystem services.

**Table 2.** Synthesis of water-related ecosystem services.

| Ecosystem Services (ES) | Ecosystem Processes |
| --- | --- |
| *Supporting ES* | |
| **Supporting the hydrological cycle** | Transpiration and evapotranspiration [44]<br>Canopy interception [45,46]<br>Hydraulic redistribution, moving water from moist to dry soil through plant roots [47,48]<br>Plants play a part in hydrological cycles by controlling water runoff [49]<br>Release of volatile organic compounds contributing to:<br>Intensification of rainfall and an overall cooling effect by blocking incoming solar energy [50];<br>Secondary organic aerosol condensing atmospheric moisture [51].<br>Trees recharge atmospheric moisture [34] and influence cloud formation [52]<br>Vegetation helps to regulate climate by cycling vast amounts of water and maintaining the gaseous composition of the atmosphere [53]<br>Terrestrial moisture recycling [54,55]<br>Precipitation recycling [34,56]<br>The biotic pump theory—precipitation in continental interiors from atmospheric circulation driven and maintained by large, continuous areas of forest starting from the coastline [57,58]<br>Arial rivers—cross-continental transport of atmospheric moisture affecting downwind water availability [34] |
| **Supporting nutrient cycling** | Forests and vegetation support biogeochemical (nutrient) cycling in four components [59]:<br>The atmosphere<br>The pool of available nutrients in the soil<br>Organic materials (living and dead)<br>Minerals in soils and rocks |
| **Supporting soil formation/quality** | Tree roots and soil organic matter from litter inputs improve soil structure, enhance aggregate stability, and promote faunal activity [36]<br>Organic matter in soil affects the saturated hydraulic conductivity by slowing down water movement [60,61] |
| **Supporting biodiversity** | Hydrology as a driver of biodiversity, supporting primary production, carrying capacity and niche formation [62,63]<br>Water as a connector linking organisms and supporting pollen and propagule dispersal [53]<br>Habitats that safeguard fisheries and biological diversity [64] |
| *Provisioning ES* | |
| **Provision of freshwater** | Tree density influence groundwater recharge [65,66]<br>Tree species influence water yield [67]<br>Tree age influence water yield [68]<br>Nutritional water productivity, i.e., 'crop per unit volume of water' [24,69,70] |
| **Provision of food and medicines** | Ecosystems provide the conditions for growing and harvesting food and extracting medicines [21] |
| **Provision of wood, fibre, and fuel** | Ecosystems provide raw materials for construction, production, and fuel, including wood, biofuels, and plant oils [21] |
| *Regulating ES* | |
| **Regulate water flow** | Water retention capacity [71,72]<br>Stream-flow regulation [73]<br>Increased infiltration from tree roots and enhanced levels of soil organic matter [36]<br>Increased infiltration capacity reduces soil evaporation losses [35]<br>Fog, mist, and cloud water capture, i.e., condensation on plant surfaces [74,75] |
| **Water purification and wastewater treatment** | Trees filter precipitation and reduce sedimentation into water courses [67]<br>Reduce pollutants entering water courses [76]<br>Natural and constructed wetlands remove pollutants [77,78]<br>Fast-growing tree species are planted to filter wastewater [79,80] |
| **Climate regulation** | Carbon sequestration in soil [81,82]<br>Carbon sequestration in above ground and below ground vegetation [83–85]<br>Regulating local temperature through evapotranspiration [86,87] |
| *Cultural ES* | |
| **Heritage value and cultural identity** | Landscape-related "memories" from past cultural ties, mainly expressed through characteristics within cultural landscapes [43] |
| **Spiritual experiences** | Holy or spiritual places important to spiritual or ritual identity, e.g., River Ganges in India, sacred forest groves, sacred plants or animals [88] |
| **Wellness, recreation and (eco)tourism** | Pleasure, comfort, discovery and socialisation that takes place in leisure in nature and observing natural elements [89] |
| **Education and research** | Climate, topography, water cycle or soil and biota used for education and research [88] |
| **Aesthetic appreciation and inspiration** | Visual perception of ecosystems and landscape [90]<br>Lakes and rivers represented in songs [91] |

## 4. Discussion

The co-production of knowledge by MSPs on water, food, forests, and landscapes has generated a number of policy and practical recommendations that help secure key water ecosystem services important for resilient landscapes and sustainable development (quadrant D, Figure 3). We examined how the knowledge production through MSPs has generated new knowledge about core governance functions, such as: coordination and management arrangements; monitoring, evaluation, learning and capacity development; participation, inclusiveness, transparency, and rule of law; and financing. We link this to a discussion of governance attributes, such as multilevel, participation, inclusiveness, transparency, impartiality, and rule of law, as discussed by Jimenez et al. (2020) [92].

### 4.1. Coordination and Management Arrangements

Multilevel governance: The MSPs on Water and Forests and the FWC both considered new perspectives related to the hydrological space within which moisture recycling takes place [34]. This has implications for management strategies and the role of regional and national governments in decision-making processes that can address the current and potential future contributions of evaporationsheds and precipitationsheds, i.e., the regional delineations of upwind locations based on thresholds of moisture contributed and received [33,56]. However, most existing water governance arrangements do not extend beyond catchments or basins to include source regions of atmospheric water production [32].

The MSPs also reviewed voluntary international commitments for landscape restoration, such as the Bonn Challenge, The New York Declaration of Forests, The Governor's Task Force, and Africa 100. In addition, national, regional, and local governments have committed to restoring hundreds of millions of hectares of wetlands worldwide. Unfortunately, progress is slow; of the 170 million hectares pledges in the New York Declaration of Forests, only 26.7 million hectares have been restored so far [93]. There is potential for these commitments to contribute to the SDG targets, and to the Paris Agreement targets. Integration of water resource management is needed for the processes to be successful in restoring resilient, productive landscape.

With respect to the UNFCCC and the Paris Agreement, the extent of national actions in the (I)NDCs intended to manage forests and landscapes for water and increased resilience, are highly interlinked with their natural potential to mitigate and adapt to climate change. However, awareness of the importance of the forest-water nexus to effectively reach the UNFCCC targets is still low [94] and the nexus is not always integrated into the (I)NDCs. The targeted review of (I)NDCs showed that forest-water synergies were rarely taken into consideration in the (I)NDC reviews with a clear sectoral focus (Figure 4). The forest-focused reviews only mentioned water once, in the context of delivering additional environmental services and extended benefits for FLR. The same sectoral division is evident in the water-focused reviews, where forest is only addressed a few times: e.g., in green adaptation measures such as reforestation to preserve groundwater; and in forest related water-conserving interventions, such as forestry management and agroforestry techniques. The lack of forest-water synergies was also evident in reviews with a general scope, where both forest and water were addressed as important factors in climate change mitigation and adaptation, but rarely linked the two sectors.

The 2020 process of enhancing the NDCs offers opportunities for countries to examine the ambition of their initial (I)NDCs and identify practical solutions for mitigating and adapting to climate change at the country level. So-called Nature-Based Solutions (NBS) are one pathway for countries to enhance climate mitigation and adaptation actions in a cost-effective manner and with multiple co-benefits [95]. NBS are intrinsically linked to landscape management, because they are inspired and supported by nature. They use or mimic natural processes to contribute to the improved management of water. Moreover, NBS, such as landscape restoration, can in some situations offer the only viable solution to urgent water management challenges. The MSP on Water and Landscape found that NBS are increasingly being mainstreamed into policies and action plans. The cost is lower

than for hard engineering measures and give multiple positive outcomes for ecosystem services and biodiversity. Challenges include the granting of land for NBS and establishing the responsibility of private individuals and state actors, to identify potential conflicts of interest [13].

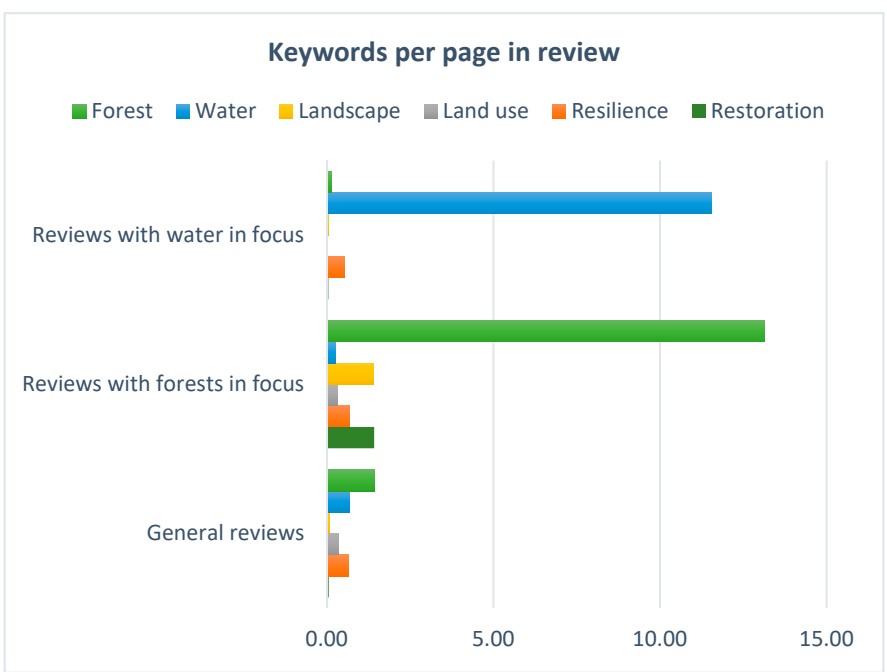

**Figure 4.** A synthesis of (I)NDC reviews illustrating the lack of integrating the forest–water nexus into the (I)NDCs.

Multi-level mechanisms, local to national, are thus involved in the governance of water in landscapes, with linkages to international commitments for, e.g., the restoration of forests and mitigation and adaptation to climate change. Through MSPs, multiple actors are brought together for dialogues on complex issues, producing knowledge at the interface between science, policy, and practice. This enables addressing difficult trade-offs between different uses of land and natural resources and to identify challenges that need urgent attention to improve core governance functions. This could be coordination and management arrangements required to support resilient landscapes that can help meet SDG6 on Water and Sanitation and other SDGs underpinned by water, such as SDG2 on Zero Hunger, SDG12 on sustainable consumption and production, SDG13 on Climate Action and SDG15 on Life on Land.

Participation, inclusiveness, transparency and rule of law: When identifying key success factors in forest landscape restoration (FLR), the MSP on Water and Forests [1,14], highlighted the importance of a conducive enabling environment, including: (1) the importance of private forest ownership and tenure, as well as the creation and development of well-organised forest owner associations and competitive forest companies; (2) transparent legislation, governance systems and regulatory frameworks, avoidance of corruption, recognised user/owner rights and clearly marked holding boundaries, impartial assessment and accurate prices in the supply chain; (3) public participation, capacity building, public awareness raising, advisory services and training in forest management, (4) integration of science and practice, including capacity building of forest owners in forest-water interactions; and (5) a prosperous forestry industry.

The MSP on Water and Food examined the importance of gender relations and found that women take different decisions regarding the selection of crops and inputs used, preferring nutritious crops as opposed to cash crops and prioritising household over irrigation water. Increasing female decision power in agriculture, access to financing, and

advisory services leads to improved agricultural productivity, and increased male and female resilience to the effects of climate change, particularly regarding the declining agricultural production and biodiversity loss [96]. Involving the youth in landscape approaches was also highlighted due to the need to create opportunities for income and employment in rural areas. There is a gap in the understanding of the role of the youth in land and water management, which is important to address when designing rural policies and programmes, and to empower and engage the youth in sustainable landscape management that continue to deliver important water-related ecosystem services. The Water and Forest MSP showed that the private sector, including smallholders, are important for improving water governance in production landscapes. By assessing water risks and addressing water in their own operations and supply chains, companies and smallholders can ensure a continuation of business and save money as operations become more effective and simultaneously, contribute to more sustainable societies [19,97].

In addition to the SDGs already mentioned above, lessons learned on the role of participation, inclusiveness, transparency, impartiality, and rule of law from MSPs related to resilient landscapes are thus relevant to SDG1 on No Poverty, SDG5 on Gender Equality, SDG10 Reduced Inequalities, and SDG16 Peace, Justice and Strong Institutions.

### 4.2. Monitoring, Evaluation, Learning and Capacity Development

The MSP on Water and Landscapes [13] concluded that the improved integration of water considerations and understanding of hydrological processes in landscapes should be part of the learning cycle, because addressing water management is often a key entry point to the restoration of degraded lands and to enhance landscape resilience and people's livelihoods. This should be coupled with the continuous development of new integrated knowledge of evidence-based management and strengthening of capacity. Monitoring frameworks are needed that integrate interactions between forests and water in the wider landscape [8]. As discussed above, strengthening multi-level governance arrangements that allow for genuine stakeholder participation in landscape management and decision making is also key. This should be accompanied by the identification and use of best management practices and innovative tools that provide practical on-the-ground solutions for sustainable management and monitoring of water in the landscape. Finally, to sustain ecosystem services important for long-term productivity, sustainability and resilience of landscapes, adequate and long-term financing from both public and private sectors need to be identified for scaling up best management practices (Figure 5).

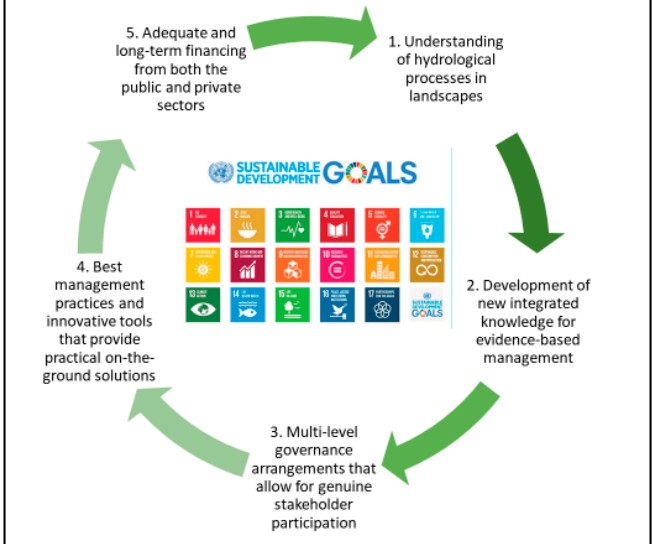

**Figure 5.** Factors to consider in restoration and sustainable management of productive landscapes.

The MSPs also concluded that bottom-up and participatory approaches to monitoring and modelling, such as citizen science, led to the inclusion of more perspectives and scenarios and more informed and comprehensive solutions. Citizen science methods are not the standard way data are collected in the field of water. Often, expensive instruments, inaccessible for local populations, are used to collect data of precipitation, streamflow, water quality and water use. Citizen science can complement other types of monitoring, but caution has to be exercised to not replace existing responsibilities of local state/government [98]. The MSP has led to collaboration with key partners in Ethiopia on capacity building in landscape restoration and monitoring of water resources using citizen science (https://www.siwi.org/what-we-do/ethiopia-water-and-landscape-governance-programme/), also using the MSP approach to engage local stakeholders in the co-production of knowledge at regional as well as local level in so called learning watersheds.

To strengthen capacity, the MSP on Water and Food demonstrated the usefulness of the concept of corporate water stewardship (CWS) that serves to unite a wide set of stakeholders in water management. A water relevance guide to standards and tools was developed (waterjourney.swedishwaterhouse.se). The tool is a stepwise, interactive process, where the first step is to understand the context in which the company operates and identify where in the value chain the company impacts water resources. The next step is to prioritise efforts and conduct feasibility analyses to provide clear definitions and outline expectations. Based on the risk mapping and assessments, policies with clear targets and indicators can be developed. The next step is to take action, and because water is often a shared resource between different users, a collective water stewardship approach is recommended. The last step is to follow up performance and to report results to stakeholders. A major challenge for food companies is to increase awareness of the water risks within the sector and to consider water in business strategies. Initiatives such as the CEO Water mandate, which is a UN Global Compact initiative, mobilises business leaders to advance water stewardship, and thereby also highlight water issues.

*4.3. Financing*

Financing of landscape restoration to ensure resilient landscapes and sustained provision of important water-related ecosystem services can be accessed from various international and national mechanisms as well as from the private sector. At the international level, this involves the financial mechanisms of the multilateral environmental agreements, including both the Green Climate Fund (GCF), the Global Environment Facility (GEF) and the Adaptation Fund (AF). At national level there is a need to promote integration both across concerned sectors and levels of governance to unlock and coordinate funding streams from key ministries and agencies [27]. Support to programmes with integrated and broad dialogue processes, such as the "Partner Driven Cooperation" developed by Sida was used as a model for bilateral forest and landscape restoration dialogues. As a result of the MSP on Water and Forests, a long-term International Training Programme on Locally Controlled Forest Restoration: A Governance and Market Oriented Approach for Resilient Landscapes has been established that includes key Swedish stakeholders from the MSP as well as a total of six countries in Africa and Asia (www.locoforest.se). The role of the private sector and smallholders in food, water and wood value-chains is crucial in ensuring sustainable production and water stewardship that contribute to the resilience of landscapes. Other innovative financing mechanisms examined by the FWC MSP include Water Funds, developed by the Nature Conservancy. They take an integrated ecosystem approach, bringing on board institutions, communities, development partners, and all relevant stakeholders.

## 5. Conclusions

In a changing environment, the drivers affecting water flows in landscapes will impact ecosystem services. Future management of landscapes will be influenced by how these drivers affect the hydrological cycle and water flows. Multiple factors combined can cause

big shifts in landscape resilience and productive capacity. Water management is critical to avoid unwanted tipping points. Water is a crucial component in most, possibly all, ecosystem services. The MSP dialogues have been useful in integrating water as part of landscape management and identifying water governance gaps, to achieve the wide range of SDGs related to water and food security, climate action, and life on land, as well as sustainable production and consumption, equality, and strong institutions.

The co-production of knowledge in the MSPs on water, food, forests, and landscapes shows that MSPs can contribute to continuous learning that informs adaptive management of water flows in landscapes, above and below ground, as well as in the atmosphere. MSPs can help to build a shared understanding of the system dynamics and integrate knowledge about hydrology and water flows into policy recommendations. They also contribute to stakeholder participation at different levels, inclusiveness and transparency, and can provide platforms for participatory and user-friendly monitoring and for strengthening stakeholder capacity and water stewardship. However, the main challenge with MSPs is to sustain interest from participants all the way from coalition building, understanding of the system, and the development of policy recommendations into collective action. Nevertheless, the MSPs discussed in this paper have acted as springboards for the mobilisation of financing and collective action in support of the water-related SDGs in Sweden and in the Global South, linking up with other water and landscape focused MSPs in partner countries.

**Supplementary Materials:** The following are available online at https://www.mdpi.com/1999-4907/12/1/1/s1, Table S1: (I)NDC reviews analysed to examine the extent of the focus on forest-water synergies.

**Author Contributions:** Conceptualisation, A.T.; methodology, A.T., L.S. and E.W.; formal analysis, A.T., L.S. and E.W.; investigation, M.G.; writing—original draft preparation, A.T., M.G., L.S. and E.W.; visualisation A.T., M.G. and E.W. All authors have read and agreed to the published version of the manuscript.

**Funding:** This research received no external funding.

**Acknowledgments:** We gratefully acknowledge support from Katarina Veem in the MSP process.

**Conflicts of Interest:** The authors declare no conflict of interest.

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
