# Peer review of "Knowledge Production for Resilient Landscapes: Experiences from Multi-Stakeholder Dialogues on Water, Food, Forests, and Landscapes"

_forests, doi:10.3390/f12010001_

Round 1

Reviewer 1 Report

This article analyses the summarized content of reports, policy briefs, scientific publications and online tools to make a general suggestion of how multi stakeholder dialogues inform governance and management of landscapes

Comments:

Section 1 –

General Comments

Clarify the objective of the paper

In what modality or platform are these discussions with stakeholders held? Is it online, in a forum, in a focus group? Who evaluates and creates these summaries that are the focus of analysis?

Specific comments

Line 49 – consider rephrasing, as the sentence form makes it appears that scientific knowledge production is prioritized and valued above diverse forms of knowledge. What are examples other forms of knowledge are incorporated into the platform? How are conflicts in knowledge mediated?

2 Materials and Methods

General Comments –

Clarify- what was the process used by the researchers to synthesize the summaries of reports and other documents? Perhaps a table summarizing types of documents analyzed sources would be helpful.

Specific Comments

line 67- a common understanding of what?

line 139 – what is a summary from an online tool?

4 Discussion

Figure 4 – If the articles reviewed identify water or forests as a primary focus, why are they included on the chart? The inclusion of these keywords seems to obfuscate the comparative representation with other keywords, as none of the other keywords register at even one tick-mark on the scale. In other words, the graphic looks distorted and there isn’t a clear reason as to why that should be the case.

  1. Results

Line 439 – This statement seems odd. There are many environmental management programs around the world in existence that focus on the water and watershed management as a scale of governance.

Examples:

Abers 2007. Organizing for governance: building collaboration in Brazilian river basins World Dev., 35 (8) (2007), pp. 1450-1463

R.N. Abers, M.E. Keck 2006. Muddy waters: the political construction of deliberative river basin governance in Brazil Int. J. Urb. Region. Res., 30 (3) (2006), pp. 601-622

Cohen 2012. Rescaling environmental governance: watersheds as boundary objects at the intersection of science, neoliberalism, and participation. Environ. Plan. A, 44 (2012), pp. 2207-2224

Cohen, S. Davidson 2011. The watershed approach: challenges, antecedents, and the transition from technical tool to governance unit. Water Altern., 4 (1) (2011), pp. 1-14

Joslin and Jepson 2018. Territory and authority of water fund payments for ecosystem services in Ecuador’s Andes. Geoforum

J.C. Rodriguez-de-Francisco, R. Boelens 2016. PES hydrosocial territories: de-territorialization and re-patterning of water control arenas in the Andean highlands Water Int., 41 (1) (2016), pp. 140-156

Author Response

Dear Reviewer,

We thank you for a very constructive review and have tried to address the issues raised to improve the manuscript. The revisions are explained point by point below and can be seen as track changes in the manuscript.

  1. Clarify the objective of the paper: We have clarified what we mean by co-production of knowledge and added a reference to this. We focus on co-production that inform the capacity to link knowledge with action in pursuit of sustainability, and have stressed that the objective of the paper is to generate policy recommendations for action (see rows 49-59).
  2. In what modality or platform are these discussions with stakeholders held: SWH MSPs are small, manageable interdisciplinary networks that meet in person at SIWI in Stockholm on a regular basis, but some partners participate via a video link if based elsewhere. The MSPs link up with ongoing international processes and networks through international meetings such as the World Water Week annual water conference in Stockholm, but there have also been meetings at other global fora outside of Sweden linked e.g. to the MEAs. (see rows 80-85)
  3. Line 49 - consider rephrasing: We have changed the wording from 'scientific and other forms of knowledge' to prodcution of knowledge from multiple sources. With the clarification of co-prodution and reference given under point 1. above, we hope that it now reads better.
  4. Materials and methods: Process used to synthesize  the summaries of reports. Perhaps a table summarizing types of documents and analysed sources would be helpful: We have added such a table to the main text that was previously found in Appendix A. As a neutral broker, SWH creates the summaries of the discussions that are reviewed and validated by all MSP members (rows 85-86). 
  5. Line 67 - a common understanding of what? Of water challenges related to sustainable development - see clarification in text.
  6. Line 139 - What is a summary from an online tool? This has been changed to ´synthesis of online tools´based on the work by the MSP on Water and Food discussed earlier in the manuscript. 
  7. Figure 4: Why are the key words water or forests included in the chart? We have rephrased the text to focus it more towards the forest-water nexus. Figure 4 is based on an analysis aimed at examining the extent to which forest-water synergies were taken into consideration in the (I)NDCs. Thus, the keywords “forest” and “water” are highly important. To compare the identified sectors, we included both reviews that had a clear sectoral division and reviews with a more general scope (see Appendix 1).
  8. Line 439 - This statement seems odd. There are many environmental management programs around the world in existence that focus on the water and watershed management as a scale of governance: We have removed this statement and rewritten the Conclusions based also on comments from Reviewer 2. The essence of the message now is that the MSP dialogues have been usefulness in integrating water as part of landscape management and identify water governance gaps.

Reviewer 2 Report

Dear Authors,

It was my pleasure to review your well-structured and informative manuscript. The literature analysis on forest-water nexus is a particularly strong aspect of your work which will surely benefit similar studies in the future. 

There are several points, however, that I recommend for further improvement:

1) Methods:

a) What is the geographical scope of your study (local, national, regional or global)? It is unclear from the text where exactly the MSPs took place? You may consider providing a map of SWH MSPs. 

b) In your paper there are two main sources of information used for analysis: dialogues and findings from different SWH MSPs + findings from FWC. What methodology did you use to combine these two main sources of data? To what extent do SWH MSPs and FWC intersect or cross-compliment in their work? Maybe there is a way to present it visually? Otherwise, the mention of FWC seems a bit confusing.

c) To what extent were the landscapes and landscape approaches present in the literature that you worked with (water-related ecosystem services and NDCs)?  

2) Co-production of knowledge:

a) How do you define the co-production of knowledge in your paper? Does it include traditional and local knowledge where applicable? How do you know that the co-production of knowledge has actually occurred as a part of SWH MSPs processes (especially if different knowledge systems were involved)? 

b) Coming back to the geographical scope of your study (and MSPs) - if it involved various regions (of one country) or even various countries, did the co-production happen in the same way? Were there any challenges or uncertainties?

3) Discussion:

I noticed that your discussion of the governance functions (4.1 to 4.3) takes more of a theoretical, global approach, however, it does not imply national or regional peculiarities of water-forest nexuses around the world (topography, ecosystem type, socio-economic characteristics, etc.). Is it the same for the Global North and Global South countries? Again, what is the role of local communities and indigenous peoples in these processes and knowledge co-production on the governance matters?

4) Conclusions:

What were the challenges encountered by MSPs? Would you recommend MSPs as a landscape approach tool across other countries and regions? Can MSPs be applied for effective management of other landscape elements (wetlands, production farmlands, coastal areas, etc.)?

5) A general suggestion: 

To further enhance your manuscript and make it a bit more empirical and a bit less conceptual, you may consider providing an example of a successful long-term MSP facilitated by your organization with its water-forest management arrangements. What worked? What didn't? What lessons can be drawn? This would substantially add value to your already great work.

I hope you find the above suggestions helpful.

Wishing you all the very best,

Reviewer

Author Response

Dear reviewer,

Thank you very much for constructive comments that have helped us to improve the clarity and quality of the manuscript. Our answers to the issues raised are summarised below:

  1. What is the geographical scope of your study (local, national, regional or global)?It is unclear from the text where exactly the MSPs took place. You may consider providing a map of SWH MSPs: We have clarified that ´SWH MSPs are small, manageable interdisciplinary networks that meet in person at SIWI in Stockholm on a regular basis, but some partners participate via a video link if based elsewhere. The MSPs link up with ongoing international processes and networks through international meetings such as the World Water Week annual water conference in Stockholm, but the MSPs have also participated in meetings at other global fora outside of Sweden linked e.g. to the MEAs. A thematic map of the MSPs also illustrating the gradual expansion of the geographical scope from Sweden to the international level and other countries is provided in Figure 2.
  2. There are two main sources of information used for analysis: dialogues and findings from different SWH MSPs+finding from FWC. What methodology did you use to combine these two main sources of data? To what extent do SWH MSP and FWC intersect or cross-compliment in their work? Maybe there is a way of present it visually? Otherwise the mention of FWC seems a bit confusing: Figure 2 illustrates the link from SWH to international water and forests networks. SWH is in turn linked to the different MSPs in Sweden (that also have some international participants). We have also moved Table 1 from the appendix to the main text where we have included several publications from the FWC that we have included in the review of documents. Recommendations from the FWC where thuis integrated into relevant steps of analysis using the conceptual framework (Figure 3).
  3. To what extent were the landscape and landscape approaches present in the literature that you worked with (water-related ecosystem services and NDCs)? Specialised research on different aspects of the hydrological cycle and ecosystem services are normally very narrow and site specific and do not discuss implications for the wider landscape, with the exception of studies on cultural ecosystem services, where landscapes are considered important for heritage values and identity.
  4. How do you define co-production of knowledge? Does it include traditional and local knowledge where applicable? How do you know that the co-production of knowledge has actually occured as part of the SWH MSP processes (especially if different knowledge systems were involved)? In the introduction we now explain that: 'Co-production is a complex concept and we therefore refrain from entering the discussion about its definitions and adopt a focus on co-production that inform the capacity to link knowledge with action in pursuit of sustainability' and we provide a reference to this discussion. Figure 2 depicts the different knowledge sources used, from small-scale farmers and forest owners, to the private sector, academia, etc. with links to interantional networks. As a neutral brooker, SWH summarised all dsicussions in writing, which was followed by review and validation by all participants.
  5. Coming back to the geographical scope of your study (and MSPs) - if it involved various regions (of one country) or even various countries, did the co-production happen in the same way? Were there any challenges and uncertainties? The MSPs all started in Sweden (see Figure 2), but gradually expanded to include international partners as well as other countries in some cases (see revised discussion). The discussion cover challenges related to funding, etc., and the conclusions mentions that a major challenge is to sustain the MSPs all the way from coalition building to collective action. Some successful examples are mentioned in the discussion, including approaches such as citizen science often used in developing countries with poor data availability.
  6. The discussion of the governance functions (4.1 to 4.3) takes a more theoretical global approach, however, it does not imply national or regional peculiarities of water-forest nexuses around the world (topography, ecosystem type, socio-economic characteristics, etc.). Is it the same for the global north and global south countries? Again, what is the role of local communities and indigenous people in these processes and knowloedge co-production on the governance matters? We have added two concrete examples to these sections. In 4.2 we explain how the MSP on water and landscape has led to collaboration with key partners in Ethiopia on capacity building in landscape restoration and monitoring of water resources using citizen science (https://www.siwi.org/what-we-do/ethiopia-water-and-landscape-governance-programme/). In section 4.3 we mention that the MSP on Water and Forests, has led to a Sida-funded a long-term International Training Programme on Locally Controlled Forest Restoration: A Governance and Market Oriented Approach for Resilient Landscapes that includes key Swedish stakeholders from the MSP as well as a total of six countries in Africa and Asia (www.locoforest.se). We cannot go into all the details of these programmes, but are providing links for further reading. A paper on the experiences of stakeholder engagement in landscape management in the Rift Valley in Ethiopia will soon be published, but cannot yet be referred to.
  7. Conclusions: What were the challenges encountered by MSPs? Would you recommend MSPs as a landscape approach tool across other countries and regions? Can MSPs be applied for effective management of other landscape elements (wetlands, production farmlands, coastal areas, etc.)? We have revised the Conclusions that are now discussing challenges with MSPs as well as opportunities, linked to the examples added in the discussion (see 6 above).
  8. A general suggestion: To further enhance your manuscript and make it more empirical and a bit less conceptual, you may consider providing an example of a successful long-term MSP facilitated by your organisation with its water-forest management arrangements. What worked? What didn't? What lessons can be drawn, etc.? As discussed above, we have added two examples where long-term knowledge production under our MSPs have led to action, which is the objective with the MSPs, not to necessarily sustain them in perpetuity. We are preparing future papers with these more concrete experiences where we will refer back to the background and theory underpinning the approach presented in this manuscript.